Protocol

# Wildfire, deforestation and health in tropical rainforest areas: a scoping review protocol

Gustavo Casais [ORCID],[1] Nathalia Sernizon Guimarães,[2] Taísa Rodrigues Cortes,[1] Julia Pescarini,[3,4] Poliana Rebouças de Magalhães,[1] Valerie Wells,[5] José Firmino de Sousa Filho,[1] Danielson Jorge Delgado Neves,[1] Michal Shimonovich,[5] Jonathan R Olsen [ORCID],[5] Edgar Marcelino de Carvalho Neto,[1] Philip Cooper [ORCID],[6,7] Srinivasa Vittal Katikireddi [ORCID],[5] Lucas Emanuel,[1,8] Roberto F S Andrade,[1,8] Gervasio Ferreira dos Santos,[1,8] Mauricio L Barreto[1,8]

For numbered affiliations see end of article.

**Correspondence to**
Dr Gustavo Casais;
gustavo.casais@hotmail.com

## ABSTRACT

**Introduction** Wildfires and deforestation potentially have direct effects on multiple health outcomes as well as indirect consequences for climate change. Tropical rainforest areas are characterised by high rainfall, humidity and temperature, and they are predominantly found in low-income and middle-income countries. This study aims to synthesise the methods, data and health outcomes reported in scientific papers on wildfires and deforestation in these locations.

**Methods and analysis** We will carry out a scoping review according to the Joanna Briggs Institute's (JBI) manual for scoping reviews and the framework proposed by Arksey and O'Malley, and Levac *et al.* The search for articles was performed on 18 August 2023, in 16 electronic databases using Medical Subject Headings terms and adaptations for each database from database inception. The search for local studies will be complemented by the manual search in the list of references of the studies selected to compose this review. We screened studies written in English, French, Portuguese and Spanish. We included quantitative studies assessing any human disease outcome, hospitalisation and vital statistics in regions of tropical rainforest. We exclude qualitative studies and quantitative studies whose outcomes do not cover those of interest. The text screening was done by two independent reviewers. Subsequently, we will tabulate the data by the origin of the data source used, the methods and the main findings on health impacts of the extracted data. The results will provide descriptive statistics, along with visual representations in diagrams and tables, complemented by narrative summaries as detailed in the JBI guidelines.

**Ethics and dissemination** The study does not require an ethical review as it is meta-research and uses published, deidentified secondary data sources. The submission of results for publication in a peer-reviewed journal and presentation at scientific and policymakers' conferences is expected.

**Study registration** Open Science Framework (https://osf.io/pnqc7/).

## STRENGTHS AND LIMITATIONS OF THIS STUDY

⇒ This scoping review will assess the health impacts of wildfires and deforestation across a broad range of tropical rainforest regions.
⇒ The search will include several different databases, including those from Latin America and Africa.
⇒ It will include manuscripts in English, French, Portuguese and Spanish, from database inception.
⇒ The ultimate selection of papers will be heavily influenced by the resolution of the shapefile used to delineate the tropical rainforest biome; this sensitivity to the map's resolution may result in the improper exclusion or inclusion of studies.
⇒ A major limitation is the absence of critical appraisal, meaning that studies with potentially low relevance, reliability, validity and applicability might be included.

## INTRODUCTION

One of the significant contributors to climate change is improper land use resulting from agriculture, logging and mining, which potentially leads to wildfires and deforestation.[1 2] Currently, wildfires and deforestation have been increasingly drawing attention for their potential consequences, not only for climate change but also for the health outcomes of both local and global populations. Governments across the world have been expressing this concern by adopting climate mitigation policies to contain environmental degradation. Understanding the health effects of wildfires and deforestation on populations in low-income and middle-income countries is critical for designing evidence-based and successful mitigation plans and policies.[3]

Tropical rainforests are home to not only a vast array of animal and plant species but also play a vital role in sustaining human well-being.[4] These ecosystems provide essential resources such as cocoa, coffee beans, bananas, vanilla and cinnamon, which are

BMJ

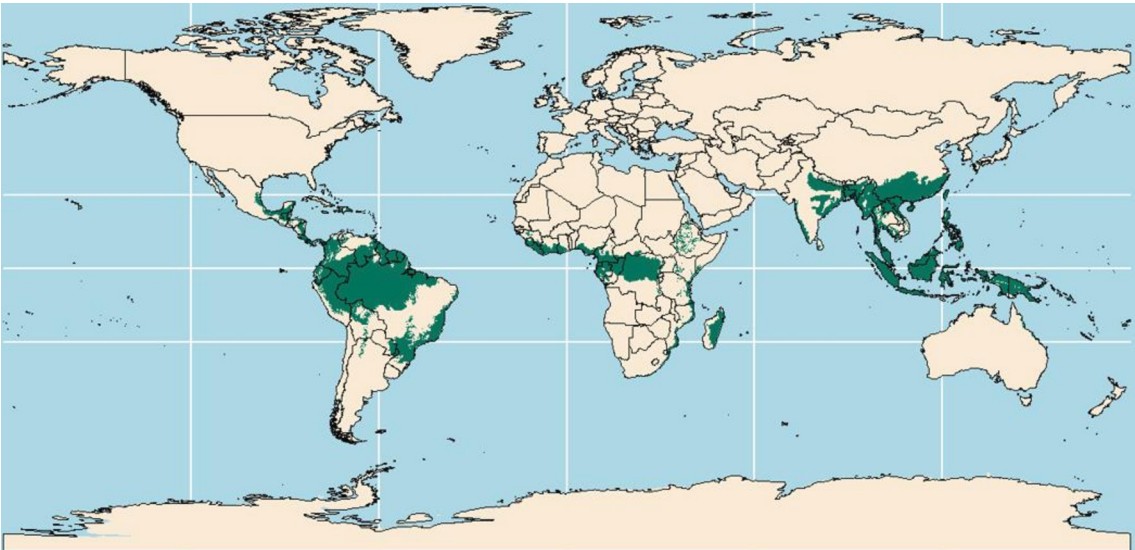

**Figure 1** Geolocation of all rainforest areas in the world.

used in everyday products. Furthermore, they are a rich source of chemical compounds instrumental in the development of medicines. These biomes are the ancestral homes of Indigenous and traditional peoples who not only live within these ecosystems but also actively preserve them along with their cultures. Rainforests are crucial in stabilising the climate and maintaining the water cycle, contributing to the overall balance of global weather patterns.[5 6] Without these rainforests, the dynamics and control of zoonotic diseases and vectorborne infections would be significantly disrupted.[7] The loss of rainforests would have severe consequences for the economy, global biodiversity and ecosystem services.[8] Figure 1 displays the location of all rainforest areas in the world according to Olson *et al*.[9]

While wildfires and deforestation often stem from common causes, they are also influenced by distinct drivers. Human activities, such as land clearing for agriculture, logging and development, play a significant role in both phenomena.[10 11] In low-income and middle-income countries, where agriculture constitutes a substantial portion of the economy, fluctuations in global demand can drive the expansion of agricultural areas, contributing to deforestation and increasing wildfire susceptibility.[12 13] Additionally, inadequate policies, weak enforcement of regulations and governance issues exacerbate both deforestation and wildfire risks by permitting unsustainable land use practices and inadequately allocating resources for fire prevention and suppression efforts.[14] Moreover, the escalating frequency and intensity of extreme events, such as droughts, extreme temperatures and storms, linked to climate change, have been amplifying wildfire occurrences globally over the past few decades.[15] Also, due to the usually high humidity in tropical rainforests, wildfires very rarely occur by natural causes, so that both deforestation and wildfires are much closely related to human activity.[16 17]

Wildfires have the potential to cause bodily injuries, impact housing infrastructure, and release toxic gases and particulate matter into the air.[18] Exposure to smoke from wildfires can cause acute respiratory illness and exacerbate existing disease, especially among children and the elderly.[19] Additionally, the long-term effects of accumulated exposures to wildfires may be multiple including premature death, cardiovascular disease, cancer, respiratory illness, mental health and other chronic conditions.[20]

Deforestation has different causal mechanisms for public health outcomes. Deforestation can alter environmental niches, changing habitats for parasites and insects, including disease-carrying mosquitoes, which may increase the human risk of contracting vectorborne diseases such as malaria and dengue.[7 21 22] Additionally, in the long run, deforestation can reduce the level of water in the atmosphere, lead to soil erosion, desertification and flooding, and increase the local temperature.[23–25]

We have noticed that previous reviews on the health impacts of wildfires and deforestation have some limitations. They often focus on specific population groups or a limited number of health outcomes, and they might not search comprehensively across databases or years.[20 26–31] Notably, no review has specifically addressed the health effects of wildfires in tropical rainforest areas, which are mainly found in low-income and middle-income countries. These regions are home to many Indigenous and marginalised groups who are particularly vulnerable due to limited resources, higher risk of health problems and limited access to healthcare.[32–34] Tropical climate, characterised by elevated temperatures, intense rainfall and high humidity, can further impact health conditions.[35 36] Moreover, the most recent reviews on the health effects of deforestation are at least 4 years old, indicating the need for an updated assessment.[30 31]

Conducting an updated review is crucial, especially considering the recent escalation of wildfires and

deforestation. This issue is particularly significant in Brazil, where fire outbreaks have been steadily increasing, even though they have not reached the peak observed in 2004.[37 38] Such a review will help identify and understand the specific ways in which wildfires and deforestation affect health outcomes, hospitalisation and vital statistics, the quantitative methods employed and the data sources used for these analyses.

Other critical areas of investigation include geographical mapping and the analytical methods employed for analysis. There is a wide variety of information sources that provide mapping for the occurrence of wildfires and deforestation, differentiating based on geographical scale and the time period of change. Disparities in high-quality environmental data around the world highlight how some places possess better information than others, which can be considered part of the phenomenon known as the 'digital divide'.[39] Additionally, various analytical methods exist to study how wildfires and deforestation impact people's health. The outcomes of such analyses heavily rely on the modelling techniques employed, underscoring the crucial need to understand and use an array of available methods. This understanding is pivotal in creating thorough and accurate insights to inform the creation of future high-quality research and understand what key research gaps exist.

This scoping review aims to comprehensively synthesise the intricate relationships between wildfires, deforestation and their impact on health outcomes in tropical rainforest regions. Our more specific objectives will be to characterise: (1) the health outcomes affected by wildfires and deforestation in the tropical areas; (2) the methods used to identify and measure their impact; (3) the data sources related to the wildfires and deforestation; and (4) the policy recommendations from the studies. This will equip policymakers and researchers with essential information about this research area, highlighting knowledge gaps and paving the way for future research and development.

## METHODS AND ANALYSIS

This scoping review protocol follows the guidelines of the Joanna Briggs Institute. The protocol is based on the framework suggested by Peters et al,[40] Arksey and O'Malley,[41] and enhanced by Levac et al.[42] It was written according to the checklist provided by the Preferred Reporting Items for Systematic Reviews and Meta-Analyses Extension for Scoping Reviews (PRISMA-ScR).[43]

The scoping review will follow the steps below: (1) identifying the research question; (2) identifying relevant studies; (3) study selection; (4) charting the data; (5) collating, summarising and reporting the results. The scoping review protocol was previously registered with the Open Science Framework to identify ongoing reviews and avoid unnecessary duplication of research.[44]

### Step 1: identifying the research question
To enhance the organisation of our research question and the criteria for inclusion and exclusion, we adhered to the mnemonic PCC (Population, Concept, Context), which is described with the research question in table 1.

According to the Oxford Dictionary, wildfire means: 'a very big fire that spreads quickly and burns natural areas like woods, forests and grassland'. We adopt this definition; therefore, we consider any large fire that occurs in different types of vegetation which can affect urban, peri-urban or rural area. The deforestation is usually associated with human activity pursuing an economic purpose (eg, farming, timber logging, expansion and infrastructure, and mining). The tropical rainforest is a warm and humid biome characterised by year-round rainfall. Renowned for its thick layers of vegetation, it consists of three distinct canopy levels located between the Tropic of Cancer and the Tropic of Capricorn.[45] We will consider the deforestation of the tropical forests located in urban, periurban or rural areas.

### Step 2: identifying relevant studies
#### Data sources
The search for scientific articles was conducted on 18 August 2023, across several databases: (1) Nursing Database (BDENF–Enfermagem), (2) National Bibliography in Argentine Health Sciences, (3) Coleciona SUS, (4) Desastres, (5) EconLit, (6) Embase, (7) Latin American and Caribbean Literature in Health Sciences, (8) Literature in Health Sciences from Caribbean countries (MedCaribe), (9) MEDLINE, (10) MEDLINE/PubMed, (11) Virtual Health Library of the Ministry of Health of Peru, (12) Literature from the Pan American Health Organization Headquarters Library, (13) Health Documentation Network in Mozambique, (14) Recursos Multimídia, (15) Scopus, (16) Western Pacific Region Index Medicus. Access to Scopus database will be via the Capes Platform, while the databases corresponding to the

| Table 1 | Scoping review questions and PCC mnemonic | | | |
|---|---|---|---|---|
| **Question** | | **Population (P)** | **Concept (C)** | **Context (C)** |
| 1. What are the impacts of the wildfire and deforestation on the health outcomes, hospitalisation and vital statistics in the tropical rainforest areas?<br>2. What are the methods and data sources used for this assessment? | | All individuals | Health impacts of wildfires and deforestation | All the areas located in the tropical rainforests |

---
**Box 1  Search terms by topics**

1. "Wildfires"[(MeSH Terms]) OR "Deforestation"[(All Fields])
2. 'All categories of human diseases' OR "Vital Statistics"[(MeSH Terms]) OR "Patient Care"[(MeSH Terms])
3. "Rainforest"[(MeSH Terms]) OR 'All country names with tropical rainforest'
4. (1) AND (2) AND (3)

The complete list of terms used can be found in the online supplemental appendix I.
---

previous identification number: (1), (2), (3), (4), (7), (8), (9), (11), (12), (13), (14) and (16) are available via the Virtual Health Library platform (Biblioteca Virtual da Saúde in Portuguese). To strengthen this review, we will also perform a manual search of the references in the included studies.

### Search strategy

The search strategy will be defined for each database, following the inclusion and exclusion criteria. The search terms were used according to the Medical Subject Headings and the respective entry terms. The expression terms were categorised into three broad aspects according to box 1: (1) the exposure, including the wildfires and the deforestation terms, (2) a comprehensive list of diseases, hospitalisation, vital statistics terms, and (3) tropical rainforest areas, including a list of countries with tropical rainforests. The countries that harbour tropical rainforests were selected according to two maps.[9 45] The expression terms will be combined with the Boolean operators 'AND' and 'OR' in the refinement. The complete search terms in every database are in the online supplemental appendix I. In the non-PubMed indexing databases, we used Emtree (for searching in Embase) and Descritores em Ciências da Saúde (for searching in the Virtual Health Library platform). Additionally, we used jargon in the search terms to ensure the inclusion of relevant local studies.

### Step 3: study selection

To ensure consistent evaluation of the literature, a two-stage screening process was employed by two reviewers. This process involved initial screening of titles and abstracts, followed by a more in-depth review of full texts. Regarding any disagreements among the authors, they were resolved either through a consensus or by the decision of one or two additional authors.

The process will be registered in a flow chart of the review process according to the PRISMA-ScR.[43] All studies were exported to the Rayyan Qatar Computing Research Institute (Rayyan), and then deduplicated by one reviewer. The partial results of the text screening are available in online supplemental appendix II.

### Inclusion criteria

To determine and choose pertinent publications concerning the topic, the subsequent inclusion criteria

will be applied: (1) quantitative studies from database inception, such as correlational, ecological, cohort, experimental and cross-sectional studies; (2) any individual or population groups exposed to wildfires, wildfire smoke or deforestation regardless of the exposure duration; (3) any disease, hospitalisation or vital statistics, considered here as birth, death rate and life expectancy; (4) self-reported health condition. Only studies written in English, French, Portuguese and Spanish will be considered for inclusion.

### Exclusion criteria

Studies will be excluded according to the following criteria: (1) theoretical studies, literature review (eg, scoping review and systematic review), letter and editorials; (2) qualitative studies (interviews, case studies, etc); (3) environmental change only (eg, extinction of wildlife, mosquitoes' habitats); (4) air pollution only (eg, air pollution from factories, mines, vehicles, without any relation to wildfires); (5) indoor fire; (6) studies solely focusing on health inequalities according to PROGRESS Plus; and (7) studies in which the exposed population is entirely outside the tropical area, as delimited by the Tropic of Cancer and the Tropic of Capricorn.

The expansive nature of this review, encompassing diverse outcomes, countries and databases, necessitates a streamlined approach. To ensure a comprehensive yet efficient exploration of the literature within the constraints of time and resources, we have opted to focus on quantitative studies. Quantitative studies often provide a more readily comparable dataset, facilitating the data extraction and the synthesis of the existing research landscape in this complex field. Additionally, the exclusion of the grey literature also relies on the constraints of time and resources.

### Step 4: charting the data

The data of interest will be extracted with the data extraction tables and by filling out the data extraction form (in the online supplemental appendix III) in Microsoft Excel. During the pilot stage, two reviewers will independently conduct the task. Afterward, one reviewer will proceed, while the work will be reviewed by a second reviewer for quality assurance. The results will be categorised according to the review questions and charted in an iterative process, allowing the reviewers to continuously update these charts when additional unforeseen data are encountered. The data extraction table will be developed and tested, containing variables on the study reference (year of publication, author, journal, full title); intervention type, exposure and data source (eg, country, origin of the exposure data source); methods and findings (study design and modelling, health outcome, control and treated group, point estimate and causal identification strategy, lag between exposure and the health consequences and limitations of the study).

## Step 5: collating, summarising and reporting the results

All gathered data will be displayed in either tabular or diagrammatic formats to visually summarise the outcomes of the studies. Initially, a table containing comprehensive information about the selected papers will be provided, such as the number of studies, study design, exposure assessment (temporal and spatial scale, and data sources), statistical methods (statistical models and identification strategies), characteristics of study populations and the countries where the studies were conducted. The data will be categorised separately for wildfires and deforestation. Different tables will be used to describe the detailed methods and data sources, followed by tables focused on findings and their subgroups. In case of studies investigating both wildfire and deforestation, we will summarise the data sources and methods used for the combined analysis and compare their effects on the outcomes. Therefore, we will synthesise the crude and adjusted effects for both analyses if available. These results will be meticulously presented, for instance, in a tabular format to underscore the collaborative analysis, as wildfires and deforestation share common contributing factors. Finally, we will consider the overall implications of the results to ensure that the scoping review will provide relevant answers to the two main research questions previously posed.

## Patient and public involvement

None.

## ETHICS AND DISSEMINATION

The study is exempt from ethics review as it is meta-research and uses published, deidentified secondary data sources. The submission of results for publication in a peer-reviewed journal and presentation at scientific and policymakers' conferences is expected.

## DISCUSSION

The resulting scoping review will offer novel insights by synthesising a wide array of health outcomes associated with wildfires and deforestation. It will elucidate the methodologies and data sources used in existing literature to assess the impact of these phenomena on public health. Moreover, the review will provide policymakers with actionable recommendations derived from studies addressing the health effects of wildfires and deforestation, including considerations of magnitude and temporal lag between exposure and outcome.

While the scoping review will offer valuable insights, it is important to acknowledge that it may encounter certain limitations. The level of detail in the map defining the tropical rainforest biome will significantly influence which studies are included in the final selection. This can lead to the unintended exclusion or inclusion of relevant research. The exclusion of the grey literature is contingent upon limitations in both time and resources available

for the study. It often contains valuable insights and data that may not be found in traditional academic sources, potentially resulting in an incomplete understanding of the topic under investigation. Additionally, the scoping review will not conduct critical appraisal. This means they cannot assess the quality of included studies, potentially incorporating biased or flawed research. Consequently, drawing definitive conclusions about the effectiveness of interventions or pinpointing areas needing strong future studies becomes difficult. While valuable for initial exploration, these limitations necessitate cautious interpretation of the findings.

**Author affiliations**
[1]Center of Data and Knowledge Integration for Health, Fiocruz/BA, Salvador, Brazil
[2]Department of Nutrition, Federal University of Minas Gerais, Belo Horizonte, Brazil
[3]London School of Hygiene & Tropical Medicine, London, UK
[4]Center of Data and Knowledge Integration for Health (CIDACS), Salvador, Brazil
[5]University of Glasgow, Glasgow, UK
[6]Universidad Internacional del Ecuador, Quito, Ecuador
[7]St George's, University of London, London, UK
[8]Federal University of Bahia, Salvador, Brazil

**Contributors** GC is the primary and corresponding author and was responsible for the first and all subsequent drafts of this scoping review protocol. Study conception—JFdSF, PC, JRO, SVK, LE, RFSA, GFdS and MLB. Design of the search strategy—JP, PRM, VW, NSG and MS. Data search—GC, JP, NSG, MS and EMdCN. Text screening—GC and TRC. Geographical map—GC and DJDN. All authors participated in discussions on the study design and critically revised drafts for improvements. ChatGPT was used as a proofreader.

**Funding** This research was funded by the NIHR (NIHR134801) using UK aid from the UK government to support global health research and by the Wellcome Trust grant (226306/Z/22/Z) awarded to the CIDACS Climate and Environmental Platform (CIDACS-Clima). JRO, MS, VW and SVK are employed by the MRC/CSO Social and Public Health Sciences Unit, University of Glasgow, and supported by the Medical Research Council (grant numbers MC_UU_00022/2; MC_UU_00022/4) and Chief Scientist Office (grant numbers SPHSU17; SPHSU19).

**Disclaimer** The views expressed in this publication are those of the author(s) and not necessarily those of the NIHR or the UK government. The researchers were independent of the funders; the funders had no role in the study design, data collection, analysis and interpretation of data, the decision to publish or the preparation of the manuscript.

**Map disclaimer** The inclusion of any map (including the depiction of any boundaries therein), or of any geographic or locational reference, does not imply the expression of any opinion whatsoever on the part of BMJ concerning the legal status of any country, territory, jurisdiction or area or of its authorities. Any such expression remains solely that of the relevant source and is not endorsed by BMJ. Maps are provided without any warranty of any kind, either express or implied.

**Competing interests** None declared.

**Patient and public involvement** Patients and/or the public were not involved in the design, or conduct, or reporting, or dissemination plans of this research.

**Patient consent for publication** Not applicable.

**Provenance and peer review** Not commissioned; externally peer reviewed.

**ORCID iDs**
Gustavo Casais http://orcid.org/0000-0002-8332-5657
Jonathan R Olsen http://orcid.org/0000-0002-5356-8615
Philip Cooper http://orcid.org/0000-0002-6770-6871
Srinivasa Vittal Katikireddi http://orcid.org/0000-0001-6593-9092

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
