## [Reviewer comments · BMJ Open]

ARTICLE DETAILS

TITLE (PROVISIONAL)	Wildfire, deforestation, and health in tropical rainforest areas: a scoping review protocol
AUTHORS	Casais, Gustavo; Guimarães, Nathalia; Cortes, Taísa; Pescarini, Julia; Rebouças de Magalhães, Poliana; Wells, Valerie; de Sousa Filho, José Firmino; Delgado Neves, Danielson; Shimonovich, Michal; Olsen, Jonathan; de Carvalho Neto, Edgar Marcelino; Cooper, Philip; Katikireddi, Srinivasa; Emanuel, Lucas; Andrade, Roberto F. S.; Ferreira dos Santos, Gervasio; Barreto, Mauricio

VERSION 1 – REVIEW

REVIEWER	Galway, Lindsay Lakehead University
REVIEW RETURNED	08-Feb-2024

GENERAL COMMENTS	Thank you for the opportunity to review this paper, I think the proposed scoping review will be an important contribution to the literature and that the protocol is, for the most part, strong. That said, I think there are some improvements to be made to both the paper and the proposed protocol. Specific comments are outlined below: ● Please revise/re-work this point “It will include manuscripts in all languages with no limitations on date. The selected papers will be sensitive to the resolution of the tropical rainforest mapping, potentially implying improper exclusion or inclusion of studies.” It is poorly written. Also, what do you mean by resolution here? Introduction: ● Overall, I think the rationale for the need and relevance of the proposed scoping review could be described more clearly and effectively. Many of the important elements are included in the introduction section but there is an absence of flow overall which makes the reader have to work harder to grasp the value of the proposal scoping review. Please review and re-organize for better flow and impact.● In both the abstract and the introduction you speak to direct and indirect impacts of wildfire smoke and deforestation. I think this distinction and discussion is important and should be aligned with existing literature (frameworks for example) and should be written about more clearly and explicitly. Moreover, should direct and indirect impacts be considered in the methods (integrated in the data charting process for example)● Consider using capital I when writing the word Indigenous● I think you should use a word other than “map” when describing the aim etc. There is likely to be some geospatial and mapping in the studies included in your review such that it is best to avoid
---

	confusion around mapping literature versus mapping with geospatial tools (characterize might be an alternative word that is more appropriate)  ● I think your aim is mischaracterized slightly and should be revised "This scoping review aims to comprehensively map the intricate relationships between wildfires, deforestation, and their impact on health outcomes in tropical regions.) This work will not be mapping the relationships between wildfire, deforestation and health. Revise to more accurately reflect what you are in fact doing. ● You have not adequately described how the scoping review research will benefit policy-makers. To do so, I would argue you should also include in your data extraction, more explicitly summary of policy recommendations discussed within papers reviewed and potentially engaged policy-makers in the scoping review process (through consultation for example, which would be inline with recommendations from Levac et al which you cite as informing your methods). I think this would greatly improve the potential relevance of your work for policy and practice. Methods and Analysis:  ● The electronic databases are comprehensive ● In terms of searching, not all cultures/contexts relevant to tropical use wildfire. This is included in the search terms in the appendix but this broader net that you are using (ie including bushfire, "wildland fires" etc.) should be noted more fully in the methods section as well as this is important given the scope of work. ● Including all languages is a strength but the methods do not outline how the team will ensure that all languages can be included. Please revise to outline this in the methods ● I suggest you calculate inter-rater reliability ● Based on your methods summary, I do not think that your team is doing a two level screening (i.e., screening title and abstract as first level and then by full text). If this is correct, I suggest that your team use a two level screening process. ● Glad to see you are considering wildfire smoke as exposure specifically ● I think you need to more fully rationalize why you are not including qualitative studies and only using quantitative data. ● Also need better rationale for why you are not including gray literature, this is increasingly expected in knowledge synthesis ● As noted above, I think your team should also aim to extract specific policy recommendations from the studies you are including. ● A final note, the author team is huge, but it appears that 3 team members are actually doing work ? Are all authors meaningful engaging in the research and/or paper writing and should be included?
--	---

REVIEWER	Chang, Andrew Y Stanford University Division of Cardiovascular Medicine
REVIEW RETURNED	01-Mar-2024

GENERAL COMMENTS	SUBMISSION SUMMARY: Casais and colleagues discuss a protocol for a scoping review of the health impacts of deforestation and wildfire in tropical rainforest regions. The authors present their systematic search protocol and plans for data analysis. GENERAL IMPRESSIONS:
--

Overall, the investigators explore an important investigative question, namely that of the impacts of key environmental hazards in tropical rainforest regions. The authors write in a clear, straightforward style and explain the importance of their proposed research. The most significant single question I have, however, is to ask what motivated the combination of deforestation and wildfire as exposures. While the two are certainly related and can be on the same causal pathway at times, they are still two distinct phenomena with overlapping but also independent etiologies. Furthermore, deforestation has a clearer human activity-linked cause and usually represents a more long-term hazard, while wildfires (acknowledging their many long-term impacts) predominantly are known for their weighty short-term impact. In the manuscript's current form, I am struggling to understand how these two exposures relate to one another in a way that necessitates their parallel inclusion in this review. Rather, should they not be explored as two separate scoping reviews?

Additionally, while the authors have provided the actual literature search terms in the Appendix section, I would appreciate more granularity explaining the search strategy in the written text, outside of the tables and appendix.

Lastly, I believe that while this submission represents a proposed study protocol, it would benefit from the inclusion of a Discussion section to contextualize the importance of the potential study and its findings. This discussion section should have a clear narrative-format Limitations subsection as well to comment on shortcomings of the proposed study design.

SPECIFIC COMMENTS BY SECTION:

*** Abstract:**

- Lines 35-36: This sentence is confusingly worded. Did the authors mean "This study will map the methods, data, and findings of the scientific literature regarding wildfire, deforestation, and health outcomes in tropical rainforests"?
- Lines 43-44: When referencing the search performed "in pairs": Do you mean "in duplicate"?
- Lines 50-51: Please specify that the work is exempt from ethics review as it is meta-research and utilizes published, deidentified secondary data sources.

*** Strengths and limitations of this study (Page 6)**

- Lines 3-4: Please clarify that the proposed scoping review will be the first meta research study (if true) to focus on assessing wildfire and deforestation impacts on health in tropical rainforest zones.
- Lines 13-14: Please specify that the lack of a critical appraisal constitutes a major limitation of the work.

*** Introduction:**

- Page 7, Lines 10-16: This summary of the existing literature requires additional citations to allow the reader to judge the veracity of the claim firsthand.
- Page 7, Lines 47-56: A figure showing the relationships between exposures, outcomes, and settings would be very helpful for the reader to understand the connections between the various factors under investigation and the aims of the authors.

	** Methods and Analysis * Step 1: Identifying the research question  - Page 8, Line 20: I would rename this subheading “Defining the research question” since the work of identifying it has already been done at the inception of the study. - Page 8, Lines 41-42: Although this is mentioned in Table 2, I would recommend including how the specific countries and geographic regions that harbor tropical rainforests were determined (i.e., if there is a certain unifying definition or resource used to decide which nations warranted inclusion or exclusion). - Page 8, Lines 43-44: As the exposures are defined in this section, I would suggest also including a description of how the health outcomes were defined. How did the investigators decide which health outcomes were worthy of inclusion and which were excluded? I noticed in the Appendix that general terms about health and medicine were not used (only disease-specific terms appear), and that mortality did not seem to have been included. ** Search strategy * Step 3: Study selection  - Page 10, Line 22-23: I am confused about the term “in pairs”. Does this mean that there are multiple teams of two? Please clarify. * Tables  - Table 2: This table could be made more informative by adding another column with examples of search terms used in each of these main concepts.
--	--

VERSION 1 – AUTHOR RESPONSE

Reviewer:

Dr. Lindsay Galway, Lakehead University

Comments to the Author:

Thank you for the opportunity to review this paper, I think the proposed scoping review will be an important contribution to the literature and that the protocol is, for the most part, strong. That said, I think there are some improvements to be made to both the paper and the proposed protocol. Specific comments are outlined below:

- Please revise/re-work this point “It will include manuscripts in all languages with no limitations on date. The selected papers will be sensitive to the resolution of the tropical rainforest mapping, potentially implying improper exclusion or inclusion of studies.” It is poorly written. Also, what do you mean by resolution here?

Authors:

We rewrote in the following manner:

- “It will include manuscripts in English, French, Portuguese and Spanish from database inception.”

- “The ultimate selection of papers will be heavily influenced by the resolution of the shapefile used to delineate the tropical rainforest biome. This sensitivity to the map’s resolution may result in the improper exclusion or inclusion of studies.”

We have revised the sentence referring to 'resolution.' The shapefile we are using may not accurately represent small portions of the forest in certain areas. As a result, there is a possibility of excluding papers due to limitations in the shapefile resolution. The shapefile depicting all rainforest areas can be seen in the figure below. Our analysis is limited to the areas between the Tropic of Cancer and the Tropic of Capricorn.

Figure 1: Rainforest shapefile

Introduction:

- Overall, I think the rationale for the need and relevance of the proposed scoping review could be described more clearly and effectively. Many of the important elements are included in the introduction section but there is an absence of flow overall which makes the reader have to work harder to grasp the value of the proposal scoping review. Please review and re-organize for better flow and impact.

Authors: Thank you for your comment. To respond to it, we rewrote most of the Introduction section.

- In both the abstract and the introduction you speak to direct and indirect impacts of wildfire smoke and deforestation. I think this distinction and discussion is important and should be aligned with existing literature (frameworks for example) and should be written about more clearly and explicitly. Moreover, should direct and indirect impacts be considered in the methods (integrated in the data charting process for example)

Authors: We have revised the text and included all of your suggestions. We enlarged the paragraphs in the introduction to stress the importance of the subject matter. Regarding the direct and indirect impacts of wildfire and deforestation, we cited them as short- and long-term impacts from wildfires and deforestation. We also included the long- and short-term effects in the Charting the Data section in the following way: “Lag between exposure and the health consequences”.

- Consider using capital I when writing the word Indigenous

Authors: Thank you. Done.

- I think you should use a word other than “map” when describing the aim etc. There is likely to be some geospatial and mapping in the studies included in your review such that it is best to avoid confusion around mapping literature versus mapping with geospatial tools (characterize might be an alternative word that is more appropriate)

Authors: You are right. We revised the word map and substitute by “synthesize” or “characterize”.

- I think your aim is mischaracterized slightly and should be revised “This scoping review aims to comprehensively map the intricate relationships between wildfires, deforestation, and their impact on health outcomes in tropical regions.) This work will not be mapping the relationships between wildfire, deforestation and health. Revise to more accurately reflect what you are in fact doing.

Authors: We think that the final paragraph of the introduction is clearly stated. Please let us know if there is any omission or part we have not addressed.

“This scoping review aims to comprehensively synthesize the intricate relationships between wildfires, deforestation, and their impact on health outcomes in tropical rainforest regions. Our more specific objectives will be to characterize: (i) the health outcomes affected by wildfires and deforestation in the tropical areas; (ii) the methods used; (iii) and the data sources related to the wildfires and deforestation; (iv) and the policy recommendations from the studies.”

- You have not adequately described how the scoping review research will benefit policy-makers. To do so, I would argue you should also include in your data extraction, more explicitly summary of policy recommendations discussed within papers reviewed and potentially engaged policy-makers in the scoping review process (through consultation for example, which would be inline with recommendations from Levac et al which you cite as informing your methods). I think this would greatly improve the potential relevance of your work for policy and practice.

Authors: Excellent suggestion. I have included this field in the Data Extraction Form. Additionally, we have included a fourth specific objective to address your comment.

“This scoping review aims to comprehensively synthesize the intricate relationships between wildfires, deforestation, and their impact on health outcomes in tropical rainforest regions. Our more specific objectives will be to characterize: (i) the health outcomes affected by wildfires and deforestation in the tropical areas; (ii) the methods used; (iii) and the data sources related to the wildfires and deforestation; (iv) and the policy recommendations from the studies.”

Methods and Analysis:

- The electronic databases are comprehensive

Authors: Thank you.

- In terms of searching, not all cultures/contexts relevant to tropical use wildfire. This is included in the search terms in the appendix but this broader net that you are using (ie including bushfire, "wildland fires" etc.) should be noted more fully in the methods section as well as this is important given the scope of work.

Authors: Bushfire, 'wildland fires,' etc., are all entry terms for the 'Wildfire' MeSH term in PubMed. We have included this information in the Search Strategy section:

"The search terms were used according to the Medical Subject Headings (MeSH) and the respective Entry Terms."

- Including all languages is a strength but the methods do not outline how the team will ensure that all languages can be included. Please revise to outline this in the methods

Authors: We updated as previously mentioned. We restricted the studies to the following languages: English, French, Portuguese and Spanish.

- I suggest you calculate inter-rater reliability

Authors: Unfortunately, we cannot fulfil that request. Despite having completed the screening process, we neglected to document the exact number of fully agreed-upon records. To the best of our recollection, out of over 6,000 screened texts, approximately 150 records were found to be in disagreement.

- Based on your methods summary, I do not think that your team is doing a two level screening (i.e., screening title and abstract as first level and then by full text). If this is correct, I suggest that your team use a two level screening process.

Authors: Actually, we performed the two-level screening. We updated with your suggested text as follows: "To ensure consistent evaluation of the literature, a two-stage screening process was employed by a team of two reviewers. This process involved initial screening of titles and abstracts, followed by a more in-depth review of full texts."

- Glad to see you are considering wildfire smoke as exposure specifically

Authors: Nice to know that.

- I think you need to more fully rationalize why you are not including qualitative studies and only using quantitative data.

- Also need better rationale for why you are not including gray literature, this is increasingly expected in knowledge synthesis

Authors: We updated the Exclusion criteria section to include the reason why we excluded qualitative studies. We wrote the following paragraph:

"To ensure a comprehensive yet efficient exploration of the literature within the constraints of time and resources, we have opted to focus on quantitative studies. The expansive nature of this review, encompassing diverse outcomes, countries, and databases, necessitates a streamlined approach. To ensure a comprehensive yet efficient exploration of the literature within the constraints of time and resources, we have opted to focus on quantitative studies.

The expansive nature of this review, encompassing diverse outcomes, countries, and databases, necessitates a streamlined approach. Quantitative studies often provide a more readily comparable data set, facilitating the data extraction and the synthesis of the existing research landscape in this complex field. Additionally, the exclusion of the grey literature also relies on the constraints of time and resources.”

- As noted above, I think your team should also aim to extract specific policy recommendations from the studies you are including.

Authors: We included this suggestion on the Data Extraction form.

- A final note, the author team is huge, but it appears that 3 team members are actually doing work? Are all authors meaningful engaging in the research and/or paper writing and should be included?

Authors: We updated Contributors section, including important tasks that were missing in the previous version:

“Contributors

GC is the primary and corresponding author and was responsible for the first and all subsequent drafts of this scoping review protocol. Study conception: PJC, JO, SVK, LE, RFSA, GFS, MLB. Designed the search strategy: JP, PBR, VW, NSG, MS. Data search: GC, JP, NSG, MS, EMCN. Geographical map: GC, DJDN. All authors participated in discussions on the study design and critically revised drafts for improvements.

Reviewer

Andrew Y Chang, Stanford University Division of Cardiovascular Medicine

Comments to the Author:

SUBMISSION SUMMARY:

Casais and colleagues discuss a protocol for a scoping review of the health impacts of deforestation and wildfire in tropical rainforest regions. The authors present their systematic search protocol and plans for data analysis.

GENERAL IMPRESSIONS:

Overall, the investigators explore an important investigative question, namely that of the impacts of key environmental hazards in tropical rainforest regions. The authors write in a clear, straightforward style and explain the importance of their proposed research. The most significant single question I have, however, is to ask what motivated the combination of deforestation and wildfire as exposures. While the two are certainly related and can be on the same causal pathway at times, they are still two distinct phenomena with overlapping but also independent etiologies. Furthermore, deforestation has a clearer human activity-linked cause and usually represents a more long-term hazard, while wildfires (acknowledging their many long-term impacts) predominantly are known for their weighty short-term impact. In the manuscript's current form, I am struggling to understand how these two exposures relate to one another in a way that necessitates their parallel inclusion in this review. Rather, should they not be explored as two separate scoping reviews?

Authors: The paper stems from the collaborative work of our team within the NIHR Unit of Social and Environmental Determinants of Health Inequalities (SEDHI). Our internal primary objective is to assess mitigation policies concerning the Amazon Forest, focusing particularly on deforestation and wildfires. With this goal in mind, we recognized the importance of conducting a scoping review to delve into the existing literature comprehensively. Given the interconnectedness of deforestation and wildfires, we concluded that a broad scoping review would not only benefit our team but also contribute to the scientific community at large.

Regarding the manner how the two phenomena are related to one another, we revised a paragraph to explain it better:

“While wildfires and deforestation often stem from common causes, they are also influenced by distinct drivers. Human activities, such as land clearing for agriculture, logging, and development, play a significant role in both phenomena [9] [10]. In low- and middle-income countries, where agriculture constitutes a substantial portion of the economy, fluctuations in global demand can drive the expansion of agricultural areas, contributing to deforestation and increasing wildfire susceptibility [11] [12]. Additionally, inadequate policies, weak enforcement of regulations, and governance issues exacerbate both deforestation and wildfire risks by permitting unsustainable land use practices and inadequately allocating resources for fire prevention and suppression efforts [13]. Moreover, the escalating frequency and intensity of extreme events, such as droughts, extreme temperatures, and storms, linked to climate change, have been amplifying wildfire occurrences globally over the past few decades [14]. Also, due to the usually high humidity in tropical rainforests, wildfires very rarely occur by natural causes, so that both deforestations and wildfires are much closely related to human activity [15] [16].”

Additionally, while the authors have provided the actual literature search terms in the Appendix section, I would appreciate more granularity explaining the search strategy in the written text, outside of the tables and appendix.

Authors: In the Search Strategy section, we added a new information so that the readers can better understand how we assembled the search strategy:

“The search terms were used according to the Medical Subject Headings (MeSH) and the respective Entry Terms.”

Besides, we think that the sentences following the excerpt compliment in satisfactory way the last statement:

“The remaining databases were adapted by Emtree and DeCS. The expression terms were categorized into three broad aspects according to Table 2: (i) the exposure, including the wildfires and the deforestation terms, (ii) a comprehensive list of diseases, hospitalization, vital statistics terms, and (iii) tropical rainforest areas, including a list of countries with tropical rainforests [40]. The expression terms will be combined with the Boolean operators ‘AND’ and ‘OR’ in the refinement. The complete search terms in every database are in the Appendix I.”

Therefore, we believe that the Search Strategy section can be well understood and replicable.

Lastly, I believe that while this submission represents a proposed study protocol, it would benefit from the inclusion of a Discussion section to contextualize the importance of the potential study and its

findings. This discussion section should have a clear narrative-format Limitations subsection as well to comment on shortcomings of the proposed study design.

Authors: Thank you for your suggestions. We included this section.

SPECIFIC COMMENTS BY SECTION:

* Abstract:

- Lines 35-36: This sentence is confusingly worded. Did the authors mean “This study will map the methods, data, and findings of the scientific literature regarding wildfire, deforestation, and health outcomes in tropical rainforests”?

Authors: We have rephrased it to make it clear.

- Lines 43-44: When referencing the search performed “in pairs”: Do you mean “in duplicate”?

Authors: We rephrased it as “The text screening was done by two independent reviewers.”

- Lines 50-51: Please specify that the work is exempt from ethics review as it is meta-research and utilizes published, deidentified secondary data sources.

Authors: Done. Thank you.

* Strengths and limitations of this study (Page 6)

- Lines 3-4: Please clarify that the proposed scoping review will be the first meta research study (if true) to focus on assessing wildfire and deforestation impacts on health in tropical rainforest zones.

Authors: Done. Thank you.

- Lines 13-14: Please specify that the lack of a critical appraisal constitutes a major limitation of the work.

Authors: Thank you. Done.

* Introduction:

- Page 7, Lines 10-16: This summary of the existing literature requires additional citations to allow the reader to judge the veracity of the claim firsthand.

Authors: We inserted three studies on the reference. Thank you for the suggestion.

- Page 7, Lines 47-56: A figure showing the relationships between exposures, outcomes, and settings would be very helpful for the reader to understand the connections between the various factors under investigation and the aims of the authors.

Authors: We have changed some words and included a (iv) specific objective. We believe that it turned the paragraph clearer.

“This scoping review aims to comprehensively synthesize the intricate relationships between wildfires, deforestation, and their impact on health outcomes in tropical rainforest regions. Our more specific objectives will be to characterize: (i) the health outcomes affected by wildfires and deforestation in the tropical areas; (ii) the methods used; (iii) the data sources related to the wildfires and deforestation; (iv) and the policy recommendations from the studies. This will equip policymakers and researchers with essential information about this research area, highlighting knowledge gaps and paving the way for future research and development.”

** Methods and Analysis

* Step 1: Identifying the research question

- Page 8, Line 20: I would rename this subheading “Defining the research question” since the work of identifying it has already been done at the inception of the study.

Authors: We appreciate your suggestion. We are following the framework suggested by Peters, et al (2015), Arksey and O’Malley (2005), and enhanced by Levac et al. (2010). The subheadings are suggested by Levac et al. (2010) and are in certain way standard in the literature.

- Page 8, Lines 41-42: Although this is mentioned in Table 2, I would recommend including how the specific countries and geographic regions that harbor tropical rainforests were determined (i.e., if there is a certain unifying definition or resource used to decide which nations warranted inclusion or exclusion).

Authors: We have added one more sentence to clarify the selection process for including the names of countries in the search terms. Terms for tropical rainforest areas were selected based on MeSH terms and Entry Terms.

“(iii) tropical rainforest areas, including a list of countries with tropical rainforests. The countries that harbor tropical rainforests were selected according to two maps [42] [43]”

- Page 8, Lines 43-44: As the exposures are defined in this section, I would suggest also including a description of how the health outcomes were defined. How did the investigators decide which health outcomes were worthy of inclusion and which were excluded? I noticed in the Appendix that general terms about health and medicine were not used (only disease-specific terms appear), and that mortality did not seem to have been included.

Authors: We have included all MeSH categories of human diseases and their respective Entry terms. For example, in the figure below, we provide a partial list of all disease categories direct from PubMed platform as an illustration of the MeSH terms of all diseases. The only category we did not include was 'Animal Diseases'. Regarding the term 'mortality,' it is included in the MeSH Term 'Vital Statistics' and its Entry terms. Therefore, using those terms should suffice to capture all subcategories. We also include a figure below to illustrate this statement for your consideration.

<https://www.ncbi.nlm.nih.gov/mesh/1000067>

** Search strategy

* Step 3: Study selection

- Page 10, Line 22-23: I am confused about the term “in pairs”. Does this mean that there are multiple teams of two? Please clarify.

Authors: Thank you for the comment. We have proofread and rewritten the paragraph:

“To ensure consistent evaluation of the literature, a two-stage screening process was employed by two reviewers. This process involved initial screening of titles and abstracts, followed by a more in-depth review of full texts. Regarding any disagreements among the authors, they were resolved either through consensus or by the decision of one or two additional authors.”

* Tables

- Table 2: This table could be made more informative by adding another column with examples of search terms used in each of these main concepts.

Authors: We have remade the table and mixed the MeSH terms with the broad categories.

Table 2 – Search terms by topics

1. "Wildfires"[MeSH Terms] OR "Deforestation"[All Fields]

- | |
|---|
| 2. 'All categories of human diseases' OR "Vital Statistics"[MeSH Terms] OR "Patient Care"[MeSH Terms] 3. "Rainforest"[MeSH Terms] OR 'All country names with tropical rainforest' 4. (1) AND (2) AND (3) |
|---|

Note: The complete list of terms used can be found in the Appendix I.

VERSION 2 – REVIEW

REVIEWER	Chang, Andrew Y Stanford University Division of Cardiovascular Medicine
REVIEW RETURNED	08-Apr-2024

GENERAL COMMENTS	GENERAL COMMENTS I appreciate the authors' clarifications of the decision to jointly examine the impacts of both deforestation and wildfires and am somewhat more convinced of their interrelatedness. Nevertheless, based on the Methods section (Page 11, Lines 32-48) it appears that these two exposures will be treated completely independently within the analysis, correct? As they lie along one another's' causal pathways, I feel that an attempt to quantitatively and/or qualitatively assess their interrelatedness in the analytic phase (e.g., discussing if any publications identified in the screening procedure jointly address both exposures) would add to the impact of this analysis. If no such analysis is possible, then it should be listed as a limitation of the work, as the Introduction section (Page 7, lines 47-49) may lead some readers to assume that the work will analyze the interaction between wildfires and deforestation as well as each exposure independently. METHODS SECTION Page 8, Lines (34-41) In the section defining tropical rainforests, I really liked the figure (the map of the tropics) the authors provided to Reviewer 1 (Dr. Galway) and would recommend inclusion of that here as a visual guide to readers less acquainted with specific rainforest regions. Page 10, Lines 45-47: I appreciate the authors' clarification of how health and vital statistics were defined. Please include the explanation given to the reviewer in the response document in the methods section to enumerate that these outcomes were defined based on the MeSH terms. As a note, such terms may not fully encompass terms for other, non-PubMed indexing databases and may represent a limitation.
---

VERSION 2 – AUTHOR RESPONSE

Reviewer: 2
 Andrew Y Chang, Stanford University Division of Cardiovascular Medicine
 Comments to the Author:

GENERAL COMMENTS

I appreciate the authors' clarifications of the decision to jointly examine the impacts of both deforestation and wildfires and am somewhat more convinced of their interrelatedness. Nevertheless, based on the Methods section (Page 11, Lines 32-48) it appears that these two exposures will be treated completely independently within the analysis, correct? As they lie along one another's' causal pathways, I feel that an attempt to quantitatively and/or qualitatively assess their interrelatedness in the analytic phase (e.g., discussing if any publications identified in the screening procedure jointly address both exposures) would add to the impact of this analysis. If no such analysis is possible, then it should be listed as a limitation of the work, as the Introduction section (Page 7, lines 47-49) may lead some readers to assume that the work will analyze the interaction between wildfires and deforestation as well as each exposure independently.

Authors:

Thank you for this important suggestion. We have identified a few papers that analyse both wildfires and deforestation on health outcomes. In the final scoping review paper, we can present how these papers addressed simultaneously both the exposures and the results. We incorporated your suggestion into the text in the following manner in the subsection 'Collating, summarising, and reporting the results':

"In case of studies investigating both wildfire and deforestation, we will summarize the data sources and methods used for the combined analysis and compare their effects on the outcomes. Therefore, we will synthesize the crude and adjusted effects for both analyses if available. These results will be meticulously presented, for instance, in tabular format to underscore the collaborative analysis, as wildfires and deforestation share common contributing factors.

METHODS SECTION

Page 8, Lines (34-41) In the section defining tropical rainforests, I really liked the figure (the map of the tropics) the authors provided to Reviewer 1 (Dr. Galway) and would recommend inclusion of that here as a visual guide to readers less acquainted with specific rainforest regions. [NOTE FROM THE EDITORS: The authors' ability to follow this request may depend on the copyright status of the figure. Is this figure reproduced from elsewhere? As BMJ Open publishes material under a Creative Commons (Open Access [OA]) licence, it can be problematic to include reproduced material, other than from another OA source, which can be reproduced under the terms of its OA licence.]

Authors:

Thank you for the suggestion. This figure was done by our team based on the data provided by Olson et al. (2001). As a result, it appears that we can confidently include it in the main text. Following this decision, we have placed the map in the introduction, directly after the addition of the following new sentence:

"Figure 1 displays the location of all rainforest areas in the world according to [9]."

Page 10, Lines 45-47: I appreciate the authors' clarification of how health and vital statistics were defined. Please include the explanation given to the reviewer in the response document in the methods section to enumerate that these outcomes were defined based on the MeSH terms. As a

note, such terms may not fully encompass terms for other, non-PubMed indexing databases and may represent a limitation.

Authors:

Thank you for the suggestion. We overlooked mentioning that we also used jargon in the original search terms. We added the following information:

“The complete search terms in every database are in the Appendix I. In the non-PubMed indexing databases, we utilized Emtree (for searching in Embase) and Descritores em Ciências da Saúde (DeCS) (for searching in the Virtual Health Library platform). Additionally, we used jargon in the search terms to ensure the inclusion of relevant local studies.”